# Dalbavancin for the Treatment of Prosthetic Joint Infections: A Narrative Review

**DOI:** 10.3390/antibiotics10060656

**Published:** 2021-05-31

**Authors:** Luis Buzón-Martín, Ines Zollner-Schwetz, Selma Tobudic, Emilia Cercenado, Jaime Lora-Tamayo

**Affiliations:** 1Department of Internal Medicine, Infectious Diseases Division, Hospital Universitario de Burgos, 09006 Burgos, Spain; 2Bone and Joint Infection Study Group of the Spanish Society of Infectious Diseases and Clinical Microbiology (GEIO-SEIMC), 28003 Madrid, Spain; sirsilverdelea@yahoo.com; 3Section of Infectious Diseases and Tropical Medicine, Department of Internal Medicine, Medical University of Graz, 8036 Graz, Austria; ines.schwetz@medunigraz.at; 4Department of Medicine I, Division of Infectious Diseases and Tropical Medicine, Medical University of Vienna, 1090 Wien, Austria; selma.tobudic@meduniwien.ac.at; 5Servicio de Microbiología y Enfermedades Infecciosas, Hospital General Universitario Gregorio Marañón, 28007 Madrid, Spain; emilia.cercenado@salud.madrid.org; 6CIBER Enfermedades Respiratorias-CIBERES (CB06/06/0058), 28029 Madrid, Spain; 7Medicine Department, School of Medicine, Universidad Complutense de Madrid, 28040 Madrid, Spain; 8Department of Internal Medicine, Hospital Universitario 12 de Octubre, Instituto de Investigación Hospital 12 de Octubre i + 12, 28041 Madrid, Spain; 9Red Española de Investigación en Patología Infecciosa (REIPI), 28029 Madrid, Spain

**Keywords:** dalbavancin, prosthetic joint infection, gram-positive

## Abstract

Dalbavancin (DAL) is a lipoglycopeptide with bactericidal activity against a very wide range of Gram-positive microorganisms. It also has unique pharmacokinetic properties, namely a prolonged half-life (around 181 h), which allows a convenient weekly dosing regimen, and good diffusion in bone tissue. These features have led to off-label use of dalbavancin in the setting of bone and joint infection, including prosthetic joint infections (PJI). In this narrative review, we go over the pharmacokinetic and pharmacodynamic characteristics of DAL, along with published in vitro and in vivo experimental models evaluating its activity against biofilm-embedded bacteria. We also examine published experience of osteoarticular infection with special attention to DAL and PJI.

## 1. Introduction

Total joint arthroplasties are common worldwide, and the incidence of this surgery is expected to increase steadily in the coming years as the population ages [1]. The most feared complication is infection, which is not associated with high mortality rates, but does carry substantial morbidity, may require many surgeries, and the final results in terms of limb functionality and pain resolution are not always satisfactory. At the same time, prosthetic joint infections (PJI) represent a massive economic burden for healthcare systems that continues to rise, and is expected to be around $1.62 billion in USA by 2030 [2]. 

PJIs are complex infections, in which the formation of biofilm, enabling bacteria to evade the host immune system, is crucial. Biofilm-embedded bacteria can also develop phenotypic changes that ultimately lead to antimicrobial tolerance and infection persistence. Not all antimicrobials perform equally in this scenario, and not all antibiotics are ideal for the treatment of PJI. In this context, the arrival of new antimicrobials is very welcome [3].

DAL is a lipoglycopeptide (Xydalba; https://www.ema.europa.eu (accessed on 27 May 2021)) that is almost universally active against Gram-positive bacteria, which are by far the leading cause of PJIs [4]. A number of clinical trials [4,5,6,7] have demonstrated its safety and efficacy for the treatment of skin and soft tissue infections, which stand as the only licensed indication for this antibiotic. However, the wide antimicrobial spectrum of this drug and its unique pharmacokinetic (PK) properties, with a half-life of 181 h [8] and prolonged concentrations in bone tissue [9], along with a good safety profile have led physicians to use it for a number of off-label indications [10,11], which include the treatment of bone and joint infections as well as PJI. In addition, resistance emergence under DAL treatment is, although possible, a very rare phenomenon. In this particular setting, the need for treatment over long periods, coupled with the long half-life of DAL mean that the antibiotic can be used on a convenient weekly basis.

In this narrative review, we assess the role of DAL in the treatment of PJI. We review the drug’s PK profile, pharmacodynamic (PD) properties, activity against biofilm-embedded bacteria in in vitro and in vivo experimental models, and finally, we evaluate the available clinical experience in PJI.

## 2. Search Strategy and Selection Criteria

The PubMed database was screened for any manuscript published at any time addressing the efficacy of DAL in the setting of biofilm-associated infections, bone and joint infections, and especially PJI. The terms “dalbavancin”, “prosthetic joint infection”, “biofilm”, “foreign-body”, “arthroplasty”, and “osteomyelitis” were combined. Abstracts and relevant full-length articles were reviewed, and a thorough search was made of the references in these papers in order to select other significant studies. Our review is not exhaustive but focuses on relevant articles regarding the efficacy on DAL on the setting of PJI, and it was restricted to articles written in English and Spanish. We directly contacted the corresponding authors of published cases series of PJI treated with DAL in order to obtain further details. 


*Definitions*


MIC_50_: Minimum inhibitory concentration required to inhibit the growth of 50% of organisms.

MIC_90_: Minimum inhibitory concentration required to inhibit the growth of 90% of organisms.

MBIC: Minimum biofilm inhibitory concentration. The lowest concentration of an antimicrobial agent required to inhibit the formation of biofilms.

MBBC: Minimum biofilm bactericidal concentration. The lowest concentration of an antimicrobial agent that eradicates 99.9% of biofilm-embedded bacteria.

## 3. Dalbavancin in Prosthetic Joint Infections

### 3.1. DAL Pharmacokinetics

The pharmacokinetics of DAL are linear and dose-proportional, with the peak concentration (C_max_) and area under the curve (AUC) increasing according to the dose administered, while its half-life (T_½_) of around 7 days remains essentially unchanged [8]. A high protein-bound fraction (>90%) contributes to this prolonged T_½_ [12,13]. It has been proven that serum bactericidal activity remains measurable at 7 days after a dose of 500 mg or higher, which establishes the basis for the weekly based dosing regimen proposed [8,14]. DAL concentrations before the following weekly dose have consistently been shown to range from 33.0 µg/mL to 40.2 µg/mL [12]. For skin and soft tissue infections, the recommended dosage consists of a loading dose of 1000 mg followed by 500 mg seven days later. C_max_ and AUC for doses of 500 and 1000 mg of DAL are 133 µg/mL and 312 µg/mL and 11,393 µg·h/mL and 27,103 µg·h/mL, respectively [8]. 

Solon et al. studied the diffusion of 20 mg/kg of DAL in bone tissue and periarticular structures by administering radioactive [^14^C]-DAL to rats. Over a 14-day period, the mean bone-to-plasma concentration ratio was 0.63, and the AUC in bone was 1125 µg eq·h/mL [15]. Later, in a phase-1 trial, Dunne et al. showed that DAL concentrations in cortical bone 12 h and 2 weeks after a single infusion of 1000 mg of DAL were 6.3 µg/g and 4.1 µg/g, respectively. In that study, the bone-to-plasma AUC ratio was determined to be 0.13. Of interest, based on population PK modeling, that study proposed a DAL regimen consisting of two 1500-mg intravenous infusions 1 week apart, which would provide concentrations in bone above the MIC_90_ for staphylococci for at least 8 weeks [9]. 

There is little information regarding intracellular concentrations of DAL. In macrophages, it has been observed to be higher than vancomycin and teicoplanin [13]. Still, we are not aware of studies on the activity against intracellular bacteria, which may be important reservoirs of infection in the setting of biofilm-associated infections [16].

In contrast with other glycopeptides (i.e., vancomycin, teicoplanin), one-third of the dose of DAL was observed to be excreted unchanged into urine, suggesting that additional non-renal pathways of elimination, probably feces, are important, as demonstrated previously in rat models [17]. Although dose adjustment does not seem necessary for mild renal impairment, patients with creatinine clearance <30 mL/min would need dose adjustment. In contrast, hemodialysis is not an important route of elimination of DAL, so that dose adjustment is not required as described in the summary of product characteristics (Xydalba; https://www.ema.europa.eu (accessed on 27 May 2021)). DAL is neither a substrate, nor an inhibitor or inducer of liver CYP-450. DAL does not require dosage adjustment in patients with hepatic impairment either [18].

### 3.2. DAL Pharmacodynamics

#### 3.2.1. Mechanism of Action and Determination of In Vitro Activity of Dalbavancin

DAL is a semisynthetic drug, structurally derived from the natural glycopeptide A40926 produced by *Nonomuraea* spp. [19], and its structure is closely related to teicoplanin. DAL inhibits the late stages of peptidoglycan synthesis interrupting bacterial cell wall synthesis by binding to the terminal D-alanyl–D-alanine terminus of pentapeptide peptidoglycan precursors [20].

Determination of DAL minimum inhibitory concentration (MIC) must be made by the standard broth microdilution method in cation-adjusted Mueller–Hinton broth supplemented with 0.002% (*v*/*v*) polysorbate-80. In addition, the gradient diffusion method procedure (Etest^®^) can be used as an alternative that has also demonstrated a high degree of agreement with the standardized broth microdilution method (EUCAST: The European Committee on Antimicrobial Susceptibility Testing. Breakpoint tables for interpretation of MICs and zone diameters. Version 11.0, 2021. http://www.eucast.org (accessed on 27 May 2021) [21]. The disk diffusion method and the agar dilution method are unreliable for the determination of susceptibility to dalbavancin.

The European Committee on Antimicrobial Susceptibility Testing (EUCAST) has defined the breakpoints for interpretation of DAL MICs only against *Staphylococcus* spp., *Streptococcus* groups A, B, C, and G, and *Streptococcus anginosus* group (*S. anginosus*, *S. intermedius*, *S.*
*constellatus*), with those isolates with DAL MICs of ≤0.125 mg/L being susceptible, and those with dalbavancin MIC values of >0.125 mg/L being resistant. In addition, EUCAST has established DAL PK/PD non-species related breakpoints, with the isolates with DAL MICs of ≤0.25 mg/L being susceptible and those with MICs > 0.25 mg/L being resistant.

#### 3.2.2. In Vitro Activity of Dalbavancin against Planktonic Gram-Positive Microorganisms

DAL is bactericidal against most Gram-positive microorganisms commonly involved in the etiology of PJI (essentially *Staphylococcus* spp., *Streptococcus* spp. and *Enterococcus* spp.).

Data from worldwide collections of strains have shown very low DAL MIC values. Of interest, the most recent data on behalf of DAL activity against all these microorganisms, as of January 2021, show that DAL MIC_90_ values have remained stable, being ≤0.06 µg/mL against different species [22]. In *S. aureus*, resistance to dalbavancin is exceptional, and the MIC_90_ is 16-fold lower than that of vancomycin (VAN) (0.06 µg/mL vs. 1 µg/mL) [23]. DAL activity has also been observed to be the same irrespective of oxacillin susceptibility [20,24], in contrast to coagulase-negative staphylococci (CoNS), which show a DAL MIC_90_ of 0.06 and 0.12 µg/mL for strains susceptible and resistant to oxacillin, respectively [25]. Since almost all *S. aureus* strains that are vancomycin-susceptible are also DAL-susceptible, vancomycin susceptibility can be considered a surrogate marker of DAL activity. Consequently, vancomycin-resistant *S. aureus* (VRSA) is also resistant to DAL, and its usefulness against heteroresistant vancomycin-intermediate *S. aureus* (hVISA) is currently a matter for debate [26]. The loss of susceptibility against other anti-Gram-positive antibiotics (i.e., teicoplanin, telavancin, daptomycin, and linezolid) does not correlate with a decrease in DAL activity [27]. In the case of CoNS, Cercenado et al. observed that DAL maintained its activity even against teicoplanin-resistant strains, as long as teicoplanin MIC was ≤8 µg/mL. (P1500: XXVII European Congress of Clinical Microbiology and Infectious Diseases; 22–25 April 2017; Vienna, Austria). In summary, according to published data, DAL is very active against *Staphylococcus* spp. with MIC_90_ values below the EUCAST susceptibility breakpoint.

Regarding enterococci, DAL activity against vancomycin-susceptible enterococci is comparable to that of staphylococci, although vancomycin-resistant *Enterococcus* spp. pose a challenge for DAL, as this antimicrobial is not active against isolates exhibiting the VanA phenotype. However, DAL is active against strains displaying the VanB phenotype (vancomycin-resistant, with variable susceptibility to teicoplanin), showing MIC_90_ values around 1 µg/mL, and it is also active against strains of *E. gallinarum* and *E. casseliflavus* that express the VanC phenotype, characterized by intrinsic resistance to vancomycin, but susceptibility to teicoplanin. Overall, it can again be assumed that vancomycin susceptibility is a good surrogate marker of DAL susceptibility in *Enterococcus* spp. and that teicoplanin susceptibility can also be used as a surrogate in vancomycin-resistant strains [23,24].

DAL activity against *Streptococcus* spp. (including penicillin-resistant *viridans* group isolates, penicillin-resistant *S. pneumoniae*, *S. anginosus* group, and ß-haemolytic streptococci) is very high. Resistance to DAL in streptococci is anecdotal, as MIC_90_ values are below 0.3 µg/mL for *S. viridans* and 0.12 µg/mL for *S. agalactiae* [23,28,29,30]. Finally, DAL has also been found to be active against other Gram-positive microorganisms eventually found to be the cause of PJI. MIC_90_ values for *Corynebacterium* spp. range between <0.03 and 0.5 µg/mL, and DAL also shows bactericidal activity against anaerobic Gram-positive cocci, such as *Peptostreptococcus* spp., *Finegoldia magna,* and *Anaerococcus* spp., with MIC_90_ ranging from 0.12 to 0.5 µg/mL [23,30,31]. Concerning its activity against *Cutibacterium acnes* (formerly *Propionibacterium acnes*), Goldstein et al. [30], in a study including 15 isolates, communicated MICs ranging from 0.03 to 0.5 mg/L, with MIC_50_ and MIC_90_ values of 0.25 and 0.5 mg/L, respectively. 

As indicated above, it is important to note that EUCAST (www.eucast.org (accessed on 27 May 27 2021)) has not defined a DAL breakpoint for *Corynebacterium* spp. and for anaerobes and defines a non-species related PK/PD breakpoint for DAL of ≤0.25 µg/mL. In this regard, antimicrobial susceptibility testing should be performed in all of the above-described organisms with MIC_90_ values of 0.5 mg/L.

#### 3.2.3. Activity of DAL against Biofilms of Gram-Positive Microorganisms: In Vitro Experience

A number of studies evaluating the activity of DAL against biofilm formation and eradication are summarized in Table 1 [30,31,32,33,34,35]. Overall, antimicrobial susceptibility studies on 96-well microtiter plates have shown that very low DAL concentrations are able to inhibit biofilm formation in a very large number of strains of staphylococci (both methicillin-susceptible and methicillin-resistant), streptococci, and enterococci (MBIC_90_ < 1 µg/mL). These values were lower than those observed for other antimicrobials such as vancomycin (MBIC_90_ 2–4 µg/mL), tedizolid, and daptomycin. Concentrations needed to eradicate biofilm are higher, with MBBC_90_ ranging from 1 to 16 µg/mL depending on the species, but they were still much lower relative to other comparators (vancomycin MBBC_90_ > 32–128 µg/mL). The exception were vancomycin-resistant enterococci, which showed very high MBIC_90_ and MBBC_90_ for all the anti-Gram-positive antimicrobials tested, including DAL. Regardless of the vancomycin resistance type (VanA or VanB phenotypes), all vancomycin-resistant enterococci had dalbavancin MICs, MBICs, and MBBCs > 16 µg/mL [33].

Information regarding the anti-biofilm activity of DAL using models with long exposure times was almost non-existent until the end of the last decade, when Di Pilato et al. evaluated the time-kill kinetics of DAL against biofilms of nine clinical strains of *S. aureus* and CoNS, using both a standardized biofilm model and biofilms grown on titanium and cobalt-chrome disks. DAL and vancomycin were used at concentrations of 1, 4, and 16 µg/mL. Against biofilms formed over 7 days on microtiter plates, the response to antibiotics was heterogeneous, although DAL showed faster and greater reduction of biofilm-embedded bacteria in the majority of the strains studied, especially at concentrations of 4 µg/mL and 16 µg/mL. In biofilms formed on Ti and Co-Cr disks, DAL was more active than vancomycin at medium concentrations (4 µg/mL), which may be expected in bone tissue [35].

More recently, Žiemytė et al. proposed a real-time, impedance-based cell analysis in order to facilitate the determination of antimicrobial susceptibility when bacteria grow in biofilms [37]. In this study, DAL ability to prevent *S. aureus* and *S. epidermidis* biofilm formation was compared with that of other antimicrobials commonly used for treating PJI (linezolid, rifampin, vancomycin, cloxacillin). The MBIC of DAL ranged from 0.5 to 2 µg/mL, and in combination with rifampin showed the highest biofilm inhibitory effect. With respect to the eradication of 6- to 9-h biofilm, DAL stopped or reduced biofilm formation at concentrations of 8–32 µg/mL. The other antimicrobials showed no activity against biofilm formed by *S. aureus*. For biofilms of *S. epidermidis*, low concentrations of DAL were active, although less than the combination of cloxacillin plus rifampin. 

#### 3.2.4. Activity of DAL against Biofilm of Gram-Positive Microorganisms: Experimental In Vivo Experience

A few in vivo experimental models [37,38,39,42] have assessed the efficacy of DAL in biofilm prevention and treatment (Table 1). Darouiche et al. compared DAL, vancomycin, and a placebo for preventing colonization of subcutaneously placed devices in a rabbit animal model inoculated with 10^3^ colony-forming units of *S. aureus*. Although not statistically significant, there was a trend toward a lower colonization rate in rabbits that received DAL before the procedure [38]. Nevertheless, the rate of foreign body contamination in rabbits receiving placebo was around 50% (lower as compared with other animal models), thus questioning the validity of the model and its discriminatory power for assessing the efficacy of antimicrobials.

In 2013, Baldoni et al. tested the ability of DAL to eliminate methicillin-resistant *S. aureus* (MRSA) biofilms in an animal model of tissue-cage infection. DAL and rifampin were administered intraperitoneally, the former at different doses (20, 40, and 80 mg/kg, which produced AUC_0–7d_ of 3393, 4298, and 4464 µg·h/mL, respectively). In monotherapy, DAL yielded a very modest killing, but in combination with rifampin, eradicated infection in one third of the cages. Of note, only the higher dosage (80 mg/kg) of DAL was able to prevent the development of rifampin resistance [39]. 

More recently, Barnea et al. studied the efficacy of DAL for the treatment of sternal osteomyelitis and mediastinitis caused by MRSA using a median sternotomy model in Lewis rats. The efficacy of DAL was proven to be similar to that of vancomycin for the treatment of sternal osteomyelitis and superior to placebo, and also reduced systemic dissemination of staphylococcal infection. DAL concentrations in bone tissue after 10 days of administration were 10.7 µg/g [40]. 

The models of animal infection suggest a role for DAL in the PJI setting, although some concerns arise after a thorough study of their results. First, in contrast to many of the in vitro studies previously reviewed, the dosages of DAL in some of the in vivo models may have provided lower antibiotic exposure compared to human PK. Second, more data on the combination of DAL with rifampin and comparisons with other rifampin-based combinations would be welcome in order to place DAL in the armamentarium of PJI caused by Gram-positive microorganisms. 

#### 3.2.5. Clinical Experience with DAL for Treating Prosthetic Joint Infections

As stated above, the broad antimicrobial spectrum of DAL and its PK properties support its use outside its approved indications. DAL is an attractive alternative in scenarios such as bloodstream infections, endocarditis, and osteomyelitis [10,43,44,45,46], even though clinical trials exploring these off-label indications of DAL are scarce.

However, in a randomized clinical trial, Rappo et al. explored the efficacy and safety of DAL for the treatment of osteomyelitis known or suspected to be caused by Gram-positive pathogens [46]. In that single-center study conducted in the Ukraine, DAL was compared with the standard of care (vancomycin was the most frequently used comparator) and the primary endpoint was clinical response at day 42. Failure was defined as the requirement of additional antibiotics, new purulence, the need for new surgery, and/or amputation. A clinical cure at day 42 was 97% in the DAL arm compared to 88% in the standard of care. Reported follow up only extends to 1 year. Even though the patients included did not have orthopedic hardware, the results are encouraging for the use of DAL in the treatment of osteitis persisting after prosthesis removal, in other words, in the setting of a two-step exchange procedure.

Meanwhile, scattered cases have been reported [47]. Furthermore, Buzón-Martín et al. reported their experience of 16 cases of PJI treated with DAL, which is so far the largest single-institution report [48]. Brief details of surgical strategies and antimicrobial treatment were provided. Overall, so as to now, 88% of patients had their infection resolved and there were no major adverse events (Buzón-Martín, unpublished data). 

In addition, a number of case series with real-world experience with DAL have been published, also including cases of PJI (Table 2) [45,48,49,50,51,52]. Common limitations found in these case series are the inclusion of small sample sizes, patient heterogeneity, aggregate outcomes of patients with PJI along with other orthopedic-related infections, and lack of details about surgical management. In fact, the goals and difficulties of treatment vary considerably depending on the type of PJI (acute vs. chronic) and whether the prosthesis is retained or removed. The main objectives of the treatment of PJI are to eradicate infection and maintain a pain-free prosthetic joint. In this context, one of three major strategies can be chosen when faced with a given PJI: To attempt eradication and cure with prosthesis retention (debridement, antibiotics, and implant retention—DAIR), attempt eradication and cure with prosthesis removal (followed by prosthesis reimplantation in either a one- or two-stage exchange procedure, or else a joint arthrodesis), or prosthesis retention, abandoning the attempt to eradicate the infection in favor of chronic suppressive antimicrobial therapy [53]. Bearing this in mind, a given antibiotic can perform very differently depending on which surgical strategy has been chosen.

An additional limitation of these studies is the wide heterogeneity in the use of DAL, even within the same institutions. Loading doses on day 1 ranged from 1000 mg to 1500 mg, and following doses at day 7 ranged from 500 to 1500 mg. The number of doses was also very variable, as some patients were treated with just two doses after prosthesis removal and others received more than 20 doses in the setting of a suppressive strategy [45,48,49,54]. Some authors [48] have even suggested that a biweekly administration strategy might be useful in this setting. As mentioned before, Dunne et al. [9] settled the rationale basis for a weekly administration of two doses of 1500 mg of DAL, and Rappo et al. proved its efficacy for treating osteomyelitis [46]. Noteworthily, these two 1500 mg doses on day 1 and day 7 of the scheme were only used in 6 out of 12 cases in the Graz series [49], but were not used in the series of Buzón-Martín, Tobudic, and Morata [45,48,50]. So far, the ideal dosing strategy of DAL for PJI remains unanswered, but perhaps two 1500 mg doses on day 1 and 7 after prosthetic removal is the scheme with a more solid investigational and clinical evidence backup [9,46].

In order to overcome the limitation of the studies heterogeneity, we contacted the authors of three of the above-mentioned case series. Dr. Zollner-Schwetz, Dr. Tobudic, and Dr. Buzón-Martín kindly provided more specific data of 36 patients treated at their institutions (Table 3). The majority of patients had already been given other antimicrobials and undergone previous surgeries, and DAL was used as salvage therapy, thus facing greater challenges. The reported etiologies were also heterogeneous, and half the patients were given DAL in combination. DAIR management was anecdotal in these cases and was only performed in two patients. The majority of infections were treated with prosthesis removal (27/36, 75%), a strategy that led to a success rate of 25/27 (92.6%) after a median follow up of 16 months. Within this group, 20/27 (74%) patients were treated with a two-stage revision procedure, two (7.4%) with single-stage revision, and three (11.1%) patients with resection arthroplasty. Of interest, a number of patients were treated with prosthesis retention plus DAL as suppressive antimicrobial therapy (7/36, 19.4%) with successful retention of the prosthesis in the short term in three cases (42.9%). Although large series of suppressive treatment with DAL for other conditions are lacking, there is some evidence to suggest that DAL can be safely administered as compassionate treatment for several months, or even years for non-surgical prosthetic endocarditis (Dr. Buzón-Martín, unpublished data). 

Overall, these revisited cases suggest that there is still insufficient experience with the use of DAL in the setting of DAIR, but that good results can be expected in the case of prosthesis removal. The use of DAL as chronic suppressive therapy could be considered in very carefully selected situations when other alternatives are lacking, although we still need more experience and information regarding the most suitable and sustainable dosage. Finally, as expected, we still need to find out which is the best DAL dosing schedule for treating PJI.

### 3.3. DAL as a Cost-Saving Strategy

Cost-saving is an additional issue, which probably justifies the use of DAL in patients with PJI. In the DALBUSE study, Bouza et al. found DAL to be cost-saving [51], and Buzón-Martín et al. observed that the use of DAL allowed an early discharge of most patients, with a presumably relevant impact in terms of healthcare costs. Applying the same cost analysis previously reported by Bouza et al. in the DALBUSE study, an estimated 571 days of hospitalization were avoided and a total of US $264,769 saved [48]. 

Several other reports position DAL as a cost-saving alternative [55,56], although, in a recent study, the results of González et al. pointed in the opposite direction [57], finding DAL to be more expensive than the standard of care for the treatment of skin and soft tissue infections. Nevertheless, in the same journal, Bookstaver et al. replied with more specific considerations other than cost and calling for other issues to be taken into account when thinking about antimicrobial stewardship [58]. It is also important to state that cost-saving analyses are quite difficult to extrapolate from the USA to other health systems in Europe, mainly those that are 100% public. 

## 4. Conclusions

DAL’s unique PK properties and high bactericidal activity are attractive characteristics for the treatment of bone and joint infections, including PJI. The possibility of using DAL in an outpatient setting, with the associated cost-saving impact, as well as the obvious improvement in therapeutical adherence compared with oral treatments, increases its value in infections where long treatments are necessary. 

With regard to this, although the specific DAL concentrations used in pre-clinical models are not always consistent with human PK, and there is very scarce information on intracellular activity, the results of DAL against biofilm-embedded bacteria are encouraging. In addition, a randomized clinical trial states that DAL is non-inferior to the standard of care in bone infections with no orthopedic hardware. The reported clinical experiences of use of DAL in PJI are scarce and heterogeneous, but its use in the setting of prosthesis removal seems reasonable and effective. We still need more data regarding its use in the setting of prosthesis retention, and also in combination with established antimicrobials such as rifampin. 

## Figures and Tables

**Table 1 antibiotics-10-00656-t001:** Summary of in vitro and in vivo pre-clinical models of dalbavancin activity against biofilm-embedded bacteria.

Reference	Microorganisms	Design	Results
Fernández et al., 2016 [32] & Schmidt-Malan et al., 2016 [34]	171 staphylococcal clinical isolates from prosthetic joint infections	Adapted Calgary-device ^1^. Biofilms were 6 h mature before confronting antibiotics during 20 h. Comparators: DAL, VAN and TDZ, at increasing concentrations.	DAL: MBIC_90_ 0.12–0.50 µg/mL, MBBC_90_ 2–4 µg/mL VAN: MBIC_90_ 2–4 µg/mL, MBBC_90_ >128 µg/mL TDZ: MBIC_90_ 2–4 µg/mL, MBBC_90_ >32 µg/mL
Knafl et al., 2017 [36]	10 MRSA plus 10 MRSE clinical strains	96-well microtiter plate with a 24 h biofilm, exposed during 24 h to increasing concentrations of DAL. Measure of remaining biofilm was made by CV dying ^2^. No comparators.	MRSA: MIC range 0.031–0.064 µg/mL; MBC 1–4 µg/mL MRSE: MIC range 0.023–0.625 µg/mL; MBC 2–16 µg/mL
Neudorfer et al., 2018 [33]	Clinical isolates 58 *E. faecalis* 25 *E. faecium*	Adapted Calgary-device ^1^. Biofilms were 6 h mature before confronting antibiotics during 20 h. Comparators: DAP and VAN	DAL: for VSE: MBIC_90_ 0.25 µg/mL, MBBC_90_ 1 µg/mL for VRE: MBIC_90_ > 16 µg/mL, MBBC_90_ >16 µg/mL VAN: for VSE: MBIC_90_ 2 µg/mL, MBBC_90_ >128 µg/mL for VRE: MBIC_90_ > 128 µg/mL, MBBC_90_ > 128 µg/mL DAP: for VSE: MBIC_90_ 4 µg/mL, MBBC_90_ 128 µg/mL for VRE: MBIC_90_ 4 µg/mL, MBBC_90_ 128 µg/mL
Di Pilato et al., 2020 [35]	9 clinical isolates plus 3 referral isolates (3 MSSA, 3 MRSA, 2 MSSE, 4 MRSE)	Model 1. Adapted Calgary device. Biofilms were 7-days mature before confronting antibiotics during other 7 d. Model 2. Ti and Cr-Co disks cultured during 48 h and then confronted to antibiotics during 7 d. Both experiments used DAL and VAN at doses of 1, 4, and 16 µg/mL	Model 1. Heterogeneous response to antibiotics. Overall, DAL showed a higher and faster reduction of biofilm-embedded bacteria over time as compared with VAN, both at lower and higher dosages. Model 2. Similar effect against biofilm formed over Ti and Cr-Co disks, except for medium dosages (4 µg/mL), where DAL showed higher reductions of biofilm-embedded bacteria
Žiemytė et al., 2020 [37]	Clinical isolates of MSSA, MRSA and MRSE	Experiments of biofilm inhibition and treatment (6–9 h-old biofilms). Measurement of biofilm growing over 20 h by electrical impedance. Treatment with increasing concentrations of DAL, CLX, VAN, LNZ, and RIF	1. Biofilm inhibition. MBIC of DAL ranged 0.5–2 µg/mL. RIF and DAL showed the highest inhibitory efficacy as compared with CLX, VAN and LNZ. 2. Biofilm treatment. DAL stopped or reduced biofilm at 8–32 µg/mL. Comparators had no effect for *S. aureus* biofilm. For *S. epidermidis* biofilm, RIF and CLX were more effective than DAL at lower concentrations.
Darouiche et al., 2005 [38]	*S. aureus* (MIC 0.06 µg/mL)	Rabbit model of infection with catheter tips implanted in subcutaneous pockets. Treatments are administered pre-operatively so to avoid the infection of the foreign material. DAL is given at 10 mg/kg, and VAN at 20 mg/kg (and then again 24 h after surgery)	In animals treated with placebo, only 47% of catheter tips were infected. The rate of infection in the DAL group was 28% (*p* = 0.2 when compared to placebo), and 53% in the VAN group (*p* = 0.8). Serum C_max_ of DAL was 80.3 µg/mL, and at day 3 it was 1.3 µg/mL. At day 7 it was only detectable in two rabbits of four (0.4 and 0.6 µg/mL).
Baldoni et al., 2013 [39]	MRSA ATCC 43300 MIC 0.078 µg/mL	Tissue-cage infection model in guinea-pigs. Treatment starts 3 days after inoculation. Three regimes of DAL: 40 mg/kg—C_max_ 44.6 µg/mL, AUC_0–7d_ 3393 µg·h/mL 60 mg/kg—C_max_ 55.6 µg/mL, AUC_0–7d_ 4298 µg·h/mL 80 mg/kg—C_max_ 68.8 µg/mL, AUC_0–7d_ 4464 µg·h/mL ^3^ T_½_ 35.8 to 45.4 h. Other regimes: DAL + RIF, RIF	DAL monotherapies had a discreet killing (inferior to RIF alone) with an infection eradication rate of 0%. The combination of RIF + DAL achieved an eradication rate similar to RIF alone (25–36%). Only high doses of DAL (80 mg/kg) avoided the emergence of rifampin resistance.
Barnea et al., 2016 [40]	MRSA (Clinical strain). MIC 0.06 µg/mL	Rat animal infection model of wound infection and sternal osteomyelitis. Treatment started 24 h after inoculation. DAL was given as an initial bolus of 20 mg/kg followed by 10 mg/kg/d for 7 or 14 days. VAN was given at 50 mg/kg/12 h for 7 or 14 days.	DAL was similar to VAN and better than the absence of treatment. Administration of DAL and VAN avoided systemic dissemination of staphylococcal infection. Concentration of DAL in bone tissue at 4, 6 and 10 days was 9.5 µg/g, 9.2 µg/g, and 10.7 µg/g, respectively

DAL: Dalbavancin. DAP: Daptomycin. VAN: Vancomycin. LNZ: Linezolid. TDZ: Tedizolid. CLX: Cloxacillin. RIF: Rifampin. MRSA: Methicillin-resistant *Staphylococcus aureus*. MSSA: Methicillin-susceptible *S. aureus*. MRSE: Methicillin-resistant *Staphylococcus epidermidis*. MSSE: Methicillin-susceptible *S. epidermidis*. VSE: Vancomycin-susceptible enterococci. VRE: Vancomycin-resistant enterococci. CV: Crystal violet. MIC: Minimal inhibitory concentration. ^1^ Pegged-lids confronted to 96-well microtiter plates. MBIC (minimal biofilm inhibitory concentration) is determined by turbidity after confronting the pegs with antibiotics. MBBC (minimal biofilm bactericidal concentration) is determined after incubating the pegged-lid in 96-well microtiter plates with fresh media after having confronted the pegs with antibiotics. ^2^ MBC was defined as a 50%-reduction in the optic density value as compared with positive controls in the 96-well microtiter plate. ^3^ PK of 80 mg/kg is comparable to data observed in blister of patients after a single dose of DAL 1000 mg (C_max_ 67 µg/mL, and AUC_0–7d_ 6438 µg·h/mL) [41].

**Table 2 antibiotics-10-00656-t002:** Clinical series published on the experience with DAL, including cases of bone and joint infection and prosthetic joint infection.

Reference	*n*	Bone & Joint Infection (Other than PJI)	Episodes of PJI	PJI Outcome (Success, %)
Bouza et al., 2017 [51]	69	13	20	80%
Morata et al., 2019 [50]	64	NP	26	NP
Tobudic et al., 2019 [45]	72	20	8	75%
Wunsch et al., 2019 [49]	101	30	32	94%
Martín et al., 2019 [48]	16	0	16	88%
Dinh et al., 2019 [52]	75	48	NP	NP

NP: not provided. PJI: prosthetic joint infection.

**Table 3 antibiotics-10-00656-t003:** Cases of PJI treated with dalbavancin according to the surgical strategy adopted (data from Buzón et al., Tobudic et al., and Wunsch et al.).

	DAIR (*n* = 2)	Prosthesis Removal (*n* = 27)	Implant Retention and Suppressive Treatment (*n* = 7)	All Patients (*n* = 36)
Sex (female)	1 (50%)	11 (40.7%)	3 (42%)	15 (43%)
Age *^,1^ (years)	69 (67–71)	69 (18–87)	62 (15–92)	67 (15–92)
Number of surgeries before DAL	1	2 (1–4)	2.5 (1–3)	1.8 (1–4)
Treatments				
DAL alone	2 (100%)	11 (40.7%)	5 (71%)	18 (50%)
DAL + rifampin	0	7 (26%)	2	9 (25%)
DAL + other treatments	0	9 (30%)	0	9 (25%)
Etiology				
*S. aureus* ^2^	0	5 (18.5%)	1 (14%)	6 (17%)
CoN staphylococci	2 (100%)	6 (22.2%)	2 (29%)	10 (28%)
*Enterococcus* spp ^3^	0	4 (14.8%)	1 (14%)	5 (14%)
Anaerobic GP	0	1 (3.7%)	0	1 (3%)
Other GP	0		2 (29%)	2 (6%)
Mixed GP	0	10 (37%)		10 (28%)
Unknown etiology	0		1 (14%)	1 (3%)
Outcome (Success)	1 (50%)	25 (93%)	4 (57%) ^4^	29 (81%)
Follow up (months) *^,1^	4 (2–6)	16 (3–40)	6 (3–14)	14 (2–40)

* Continuous variables are expressed as median and (range). ^1^ Data available for 20 patients. ^2^ There were 4 methicillin-susceptible strains (3 managed with prosthesis removal and 1 by suppressive antimicrobial therapy), and 2 methicillin-resistant strains (both managed by prosthesis removal). ^3^ There were 4 *E. faecium* (all treated with prosthesis removal) and 1 *E. faecalis* (treated with suppressive antimicrobial treatment). ^4^ One patient died to unrelated causes after three months with no clinical or biochemical signs of failure. Abbreviations: DAL: dalbavancin. CoN: coagulase-negative. GP: Gram-positives. DAIR: debridement, antibiotics, and implant retention.

## Data Availability

Not applicable.

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
