# Peer review of "Dalbavancin for the Treatment of Prosthetic Joint Infections: A Narrative Review"

_antibiotics, 2021, doi:10.3390/antibiotics10060656_

Round 1

Reviewer 1 Report

Authors reviewed all publications about dalbavacin, which is an interesting subject for multidisciplinary team dealing with PJI. Nevertheless, I find the paper very difficult to follow. Authors should be more synthetised. They mainly resume each publication and report corresponding information.

General comments :

Too many abreviations and VAN abreviation for vancomycin is confusing with the VAN for phenotype resistance

Et al in italics . i

t is better to have a line spacing

Abstract :

Last sentence : Redundant with previous sentence plus this part of reviews lack significant conclusions. I suggest to only speak about clinical outcome regardless surgical treatment .

Introduction :

Line 50 : need a reference

Line 58 : reference needed

Line 68 : ref needed

The major advantage of dalbavancin is the long half-life. It is essentiel to precise the half-life in abstract and introduction for readers. Not all specialists.

In the introduction, authors speak a lot about impact on biofilm and neglect other major points to take in consideration in PJI management and antibiotics choice : tolerance for a long duration of treatment , bone penetration, emergence of resistance.

Line 76-78 : what s the point ?

Line 83 : clinical experience. Clinical experience in PJI.

Search strategy and selection criteria

This section, the basis of following information, need to be rewritten to include more information as period of search , information collected , number of publications in each following section (pk,pd,cases)

Dalbavacin in PJI

Dal pharmacokinetics

A figure corresponding to information in the first paragraph would be interesting to follow. More information about methods in the studies is needed.

Second paragraph :

Precise dose infused for both studies.

Report more information about why and based and what a regimen with two infusions was proposed. Discuss the point more extensively in PJI section.

Authors didnot mention in the paper what is recommended posology for dalbavancin .

Remove last sentence in this second paragraph.

Line 142 : against ?

Line 151 . website should be included in the references section. Information in the summary of the product should be briefly reported in introduction section to expose clearly to readers the molecule.

Dal pharmacodynamics

Subtitles for each section should be rewritten e.g first one in vitro planktonic

Breakpoint should be reported at the beginning of the section.

It is very difficult to follow this section with lots of values and abreviation :

A table summary for all reported values for different species with eucast breakpoints is needed.

Technique used to measure MIC shoud be reported .

For Staphylococus, separate in the text S.aureus and CoNS information.

Biofilm results

Definition of MBBC and MBIC should be exposed before showing results. Maybe a definition section with definition of MIC50,MIC90,MBBC and MBIC… will be helpful

Lines 157-159 : remove this sentence

Lines 272-284 : should be completely rewritten. We don’t understand on with species it was tested. Sometimes , values are reported sometimes they are not .

In vivo experience

Line 298 : why 50% i slow ?

Real world experience

This title should be in italics ?

Lines 332 : clinical than « real-world »

Report what are adverses events

The title of this paragraph is experience in PJI and in the paragraphs all infections were included. I suggest to include only PJI data and at the beginning of the section report briefly number of clinical publications. This section should be summarized . too much information not concerning PJI . authors propose in the abstract to revisit publications to adress dalbavancin efficacy in light of surgical management but in this section all clinical data was reported without any selection . it is impossible to follow and make conclusions

Paragraph starting Lines 433 : Precise for how many publications you were able to have more data

Table 1 : more columns are need . e.g design split into : model of biofilm and comparator . results split into MBIC/MBBC/antibiotic tested

Table 2 is entitled table 1 . define GP

Author Response

We attach the answers to Reviewer´s suggestions.

Thank you very much

Reviewer 2 Report

This is a very well written paper involving an extensive literature review explaining the PK/PD data surrounding the drug dalbavancin, its recorded efficacy in the setting of biofilm presence, and its potential efficacy in PJI (albeit this data is limited). The authors do a great job in assimilating the available data, coming up with conclusions and determining shortcomings that require further investigation. It is excellent as written and I have no major or minor adjustments I would make to this paper--would accept as is! Well done!

Author Response

Reviewer 2: This is a very well written paper involving an extensive literature review explaining the PK/PD data surrounding the drug dalbavancin, its recorded efficacy in the setting of biofilm presence, and its potential efficacy in PJI (albeit this data is limited). The authors do a great job in assimilating the available data, coming up with conclusions and determining shortcomings that require further investigation. It is excellent as written and I have no major or minor adjustments I would make to this paper--would accept as is! Well done!

We are thankful to the reviewer for his very kind words, we really appreciate them.

Reviewer 3 Report

Buzón-Martínez al present a narrative review of dalbavancin, focused in prosthetic joint infections. This indication is not well studied and in my opinion the review is of interest. It is well written and easy to read. I suggest some minor comments to improve the final version. 

Minor comments:

Line 63. To my knowledge, dalbavancin was commercialized on 2014 and there have been several new antibiotics since then. Consider changing the adjective “recently” for the year of commercialization.

Line 87. Precise the data search for publications.  

Line 342. It seems that the reference 42 is not in the text.  

Author Response

Estimated Sirs,

I would appreciate if the MANUSCRIPT PDF could be sent to reviewer number one as I have not sent it by mistake.

Reviewer 3: Buzón-Martín al present a narrative review of dalbavancin, focused in prosthetic joint infections. This indication is not well studied and in my opinion the review is of interest. It is well written and easy to read. I suggest some minor comments to improve the final version. 

Minor comments:

Line 63. To my knowledge, dalbavancin was commercialized on 2014 and there have been several new antibiotics since then. Consider changing the adjective “recently” for the year of commercialization. We agree with the reviewer, dalbavancin is no longer such a recently commercialized drug, time flies indeed. We have corrected line 63”, accordingly.

Line 87. Precise the data search for publication. Given the scarcity of publications addressing the efficacy of DAL for prosthetic joint infections and biofilm-associated infections, we have included in our first search any publication at any time. We have modified the text accordingly.

Line 342. It seems that the reference 42 is not in the text. We thank the reviewer for giving us the opportunity of amending this error. We have introduced ref 44, line 392.

Round 2

Reviewer 1 Report

Authors made a significant work to improve the quality of the review and make it  of a better impact to readers .